# Water Deficit at Vegetative Stage Induces Tolerance to High Temperature during Anthesis in Rice

**DOI:** 10.3390/plants12173133

**Published:** 2023-08-31

**Authors:** Anderson da Rosa Feijó, Vívian Ebeling Viana, Andrisa Balbinot, Marcus Vinicius Fipke, Gustavo Maia Souza, Luciano do Amarante, Luis Antonio de Avila

**Affiliations:** 1Plant Physiology Graduate Program, Federal University of Pelotas, Pelotas 96160-000, Brazil; anderson.feijo@ufpel.edu.br (A.d.R.F.); gmsouza@ufpel.edu.br (G.M.S.); luciano_amarante@ufpel.edu.br (L.d.A.); 2Crop Protection Graduate Program, Federal University of Pelotas, Pelotas 96015-560, Brazil; vivian.viana@ufpel.edu.br (V.E.V.); abalbinot@ufpel.edu.br (A.B.); mfipke.faem@ufpel.edu.br (M.V.F.); 3Department of Soil and Crop Sciences, Mississippi State University, Mississippi State, MS 39762, USA

**Keywords:** *Oryza sativa* L., climate changes, CO_2_, priming, yield, molecular markers

## Abstract

Background: Crop yields have been affected by many different biotic and abiotic factors. Generally, plants experience more than one stress during their life cycle, and plants can tolerate multiple stresses and develop cross-tolerance. The expected rise in atmospheric CO_2_ concentration ([CO_2_]) can contribute to cross-tolerance. Priming is a strategy to increase yield or to maintain yield under stress conditions. Thus, our objective was to evaluate if priming the rice plants with water deficit during the vegetative stage can induce tolerance to heat stress at anthesis and to evaluate the contribution of *e*[CO_2_]. Methods: The experiment was arranged in a completely randomized design in a factorial arrangement. Factor A consisted of the following treatments: water deficit at four-leaf stage (no-stress, and drought stress), heat at anthesis (normal temperature, high temperature), and priming with water deficit at four-leaf stage and heat stress at anthesis; and Factor B was two [CO_2_] treatments: *a*[CO_2_] = 400 ± 40 μmol mol^−1^ and *e*[CO_2_] = 700 ± 40 μmol mol^−1^. We assessed the effect of the treatments on plant growth, yield, biochemical, and transcriptome alterations. Results: Although *e*[CO_2_] affected rice growth parameters, it did not affect the priming effect. Primed plants showed an increase in yield and number of panicles per plant. Primed plants showed upregulation of *OsHSP16.9A*, *OsHSP70.1*, and *OsHSP70.6*. These results showed induced cross-tolerance. Conclusions: Water deficit at the rice vegetative stage reduces the effect of heat stress at the reproductive stage. Water deficit at the vegetative stage can be used, after further testing in field conditions, to reduce the effect of heat stress during flowering in rice.

## 1. Introduction

Climate change can affect plant growth and yield. On one hand, enhanced CO_2_ concentration may positively affect plant growth, especially C_3_ plants; on the other, enhanced temperature and other environmental stress can negatively affect grain yield. With climate change, environmental stresses such as water deficit and heat stress are becoming more frequent. The stresses caused by water deficit and high temperatures (heat) are the most important environmental stresses that have strongly impacted world agriculture, mainly because they affect the growth and productivity of the main crops [1,2,3]. Rice (*Oryza sativa* L.) is susceptible to water deficit and high temperatures at all development stages, especially during the reproductive stage. Depending on the severity of the stresses, the effects are inhibition of growth and development, reduced grain yield, and, in more severe cases, plant death [4,5,6,7]. Under natural conditions, the combination of one or more stresses, such as the occurrence of water deficit and extreme temperatures, are common in many agricultural areas [8]. The severity and the frequency of these combined stresses is increased due to climate change [9]. In southern Brazil, a rice producer uses a form of rice management that is now called by his surname, Forsin, which involves stressing the rice plant at the vegetative state followed by the flooding of the area, obtaining steady high yield [10]. His management drove our curiosity, and our hypothesis was not only that his soil and rice nutrition was very high, but also that he is empirically using the drought to prime his rice and to overcome further stresses.

Plants are exposed to two or more stressors simultaneously, and can also experience different stress conditions sequentially during their cycle being affected by cross-stresses. Recently, the study of the phenomenon of cross-tolerance between stresses has gained more attention. This phenomenon is characterized by the acquisition of tolerance to a second, more severe stress, after previous exposure to a primary stress of a different nature and lower intensity [11,12]. Exposure to primary stress triggers a process known as priming. The priming effect induces a memory of stress through changes in plant metabolism at the biochemical, molecular, and epigenetic levels [13]. These alterations are activated mainly via the ROS-regulated signaling cascade which, at low-to-moderate concentrations, acts as secondary messengers, with emphasis on the action of hydrogen peroxide (H_2_O_2_), which acts by modulating the expression of genes involved in tolerance to various stresses [11,14].

The occurrence of cross-tolerance between water deficit and heat stress has been reported in different plant species such as *Festuca arundinacea*, wheat, and *Poa pratensis* [15,16,17]. However, the studies developed to assess the occurrence of the cross-tolerance phenomenon, in general, have not considered the expected increase in atmospheric [CO_2_]. In this context, our hypothesis is as follows: (1) rice plants develop cross-tolerance to heat stress at the flowering stage after being previously exposed to a period of water deficit at the V-4 stage, inducing an increase in system activity enzymatic antioxidant and HSPs; and (2) the increase in atmospheric [CO_2_] concentration favors the development of this type of cross-tolerance, inducing greater efficiency of the enzymatic antioxidant system and of the activity of HSPs in relation to the ambient CO_2_ condition. 

Therefore, to verify the possible development of cross-tolerance between stresses, the objectives of this study were to evaluate if priming the rice plants with water deficit during the vegetative stage can induce tolerance to heat stress at anthesis through morphological, biochemical, and molecular responses in rice plants, and furthermore, to evaluate the contribution of increasing atmospheric [CO_2_] concentration on the occurrence of cross-stress.

## 2. Results

### 2.1. Analysis of Variance

The results of the analysis of variance for each factor, CO_2_ concentration ([CO_2_]) and stress caused by water deficit, heat stress, and priming, and their interactions with the parameters analyzed in the experiment are presented in Table 1. Changes in shoot and root mass weight were not detected as the effect of the increase in [CO_2_] nor the priming, water deficit (WD), and heat (HT) treatments. The H_2_O_2_ content, SOD and CAT activities, and chlorophyll a and b and its ratio were not affected in plants that received priming or in plants that were only treated with WD or HT showing only the effect of [CO_2_]. Changes in ascorbate peroxidase (APX) activity and carotenoid content were not detected as the effect of the increase in [CO_2_] nor the priming, WD, or HT treatments. Thus, the increase in [CO_2_] did not affect cross-tolerance development in our experimental conditions.

### 2.2. Effect on Rice Yield Components

As no interaction was detected related to increasing in [CO_2_] and priming treatment, we show separately the simple effects of the [CO_2_] and priming treatments in the yield components. Elevated CO_2_ concentration *(e*[CO_2_]) affected the yield components (Figure 1). The number of panicles per plant, thousand-grain weight, and yield were positively affected by the *e*[CO_2_] (Figure 1A,D,E). The *e*[CO_2_] decreased spikelet sterility compared to plants grown under current ambient CO_2_ concentration (*a*[CO_2_]) (Figure 1B). On the other hand, the number of grains per panicle was negatively affected by *e*[CO_2_] (Figure 1C).

Priming with water deficit at the vegetative stage followed by heat stress (WD + HT) during anthesis positively affected the number of grains per panicle, obtaining higher values compared to all other treatments, including the control condition (Figure 2A). For rice grain yield (Figure 2C), WD + HT produced a higher yield compared to the naïve plant (HT), which received only heat stress, and to the primed without the second stress (WD); no difference was observed between WD + HT and the control condition, demonstrating that the plants that were primed could overcome the heat stress better than the naïve plant. The weight of thousand-grain was not significantly affected by the stress conditions when compared to the control condition (Figure 2B); the only treatment that had a higher thousand grain weight was the HT.

### 2.3. Effects on Photosynthesis and the Antioxidant Mechanism

Following the ANOVA (Table 1), H_2_O_2_, SOD, CAT, and chlorophyll a and b and its ratio were affected by [CO_2_], with no significant differences in priming treatment. The *e*[CO_2_] greatly affected rice cell homeostasis. The hydrogen peroxide (H_2_O_2_) was significantly increased in *e*[CO_2_] (Figure 3A). Consistently, the activities of superoxide dismutase (SOD) and catalase (CAT) were greatly increased in *e*[CO_2_] (Figure 3B,C). Chlorophyll *a* and *b* were significantly reduced at *e*[CO_2_], but the ratio between them was not affected by *e*[CO_2_] (Figure 3D–F).

Priming only affected the malonyldialdehyde (MDA) accumulation (Figure 4) that showed higher values in the primed plant WD + HT compared to the plant that only received WD. The values did not differ from the control condition and the HT. 

### 2.4. Effects on HSP Transcriptional Regulation

Heat shock proteins (HSPs) are the main molecular makers of stress responses such as heat and water deficit, and here, we identified that they are also differentially transcribed in response to changes in [CO_2_] (Figure 5). *e*[CO_2_] modulated the transcriptional regulation of all tested *OsHSPs* in rice flag leaves except for *OsHSP70CP1* (0.11-fold), which was not differentially regulated by *e*[CO_2_] under WD conditions, and *OsHSP20* (−0.21-fold), which was not differentially regulated by *e*[CO_2_] under HT (Figure 5A). Interestingly, *OsHSP16.9* (4.9-fold HT; −0.98-fold WD), *OsHSP24.1* (−2.7-fold HT; 1.0-fold WD), *OsHSP17.9A* (0.5-fold HT; 1.1-fold WD), *OsHSP20* (−0.2-fold HT; 1.3-fold WD), *OsHSP70.1* (3.7-fold HT; 2.4-fold WD), and *OsHSP70.6* (3.8-fold HT; 2.6-fold WD) were upregulated in response to WD or HT. However, all of them, *OsHSP16.9, OsHSP24.1, OsHSP17.9A*, *OsHSP20, OsHSP70.1*, and *OsHSP70.6*, were downregulated in WD + HT as an effect of *e*[CO_2_] showing −4.2, −0.3, −2.4, −2.2, −3.2, and −2.5-fold, respectively. It shows that *e*[CO_2_] displayed a negative effect downregulating HSP expression in rice plants (Figure 5A).

Differences between primed flag leaves in *a*[CO_2_] and *e*[CO_2_] were not observed (Figure 5B). The priming effect was negatively affected by *e*[CO_2_]. *OsHSP16.9* (1.2-fold WD + HT; 6.1-fold HT; 1.5-fold WD), *OsHSP20* (0.3-fold WD + HT; 1.5-fold HT; −0.5-fold WD), *OsHSP70.1* (0.6-fold WD + HT; 2.6-fold HT; 2.0-fold WD), and *OsHSP70.6* (−0.3-fold WD + HT; 3.5-fold HT; 1.2-fold WD) were downregulated after priming in *e*[CO_2_] with respect to its expression in HT or WD (Figure 5B). In *e*[CO_2_], the priming effect was observed only for *OsHSP24.1* in *e*[CO_2_] that showed an increase in transcript accumulation in WD + HT by 1.7-fold, with respect to −0.45 and −0.1-fold in HT and WD, respectively. On the other hand, the priming effect was observed in flag leaves in *a*[CO_2_]. *OsHSP16.9* (4.2-fold WD + HT; 1.3-fold HT; 0.7-fold WD), *OsHSP70CP1* (1.1-fold WD + HT; 0.3-fold HT; −0.4-fold WD), and *OsHSP17.9* (1.4-fold WD + HT; 0.8-fold HT; −1.1-fold WD) were upregulated after priming in *a*[CO_2_] with respect to its expression in HT and/or WD (Figure 5B).

*OsHSPs* were also responsive to *e*[CO_2_] in rice spikelets (Figure 6A). The *e*[CO_2_] modulated the transcriptional regulation of *OsHSPs* in rice spikelet with a lesser effect than that observed in rice flag leaves. Different from that observed in flag leaves, the transcriptional regulation of *OsHSPs* (*OsHSP16.9A*, *OsHSP17.9*, and *OsHSP70.6*) was greatly affected by the *e*[CO_2_] since they were upregulated 4.6-, 0.5-, and 0.6-fold, respectively, at the control condition. *OsHSP16.9A* and *OsHSP17.9A* were upregulated in HT as function of *e*[CO_2_]. *OsHSP70.1* (1.9-fold WD; −4.0-fold HT) and *OsHSP70.6* (3.2-fold WD; −4.9-fold HT) were upregulated in WD by the influence of *e*[CO_2_] while were strongly downregulated in HT due the *e*[CO_2_] effect. Different from the observed for primed flag leaves, *e*[CO_2_] helped the spikelet priming. *OsHPSP70.1* (2.3-fold WD + HT; −4.0 HT; 1.9-fold WD; 0.2-fold CNT) and *OsHSP70.6* (3.6-fold WD + HT; −4.9-fold HT; 3.3-fold WD; 0.6 CNT) were upregulated in primed plants with respect to the control condition and the HT and WD conditions, showing that increased [CO_2_] greatly affected the priming effect in rice spikelets. In primed spikelets, *OsHSP24.1* (−1.0-fold), *OsHSP70CP1* (−2.3-fold), and *OsHSP20* (−0.4-fold) were negatively affected by *e*[CO_2_].

The priming with WD in rice spikelets showed to be effective due to the great transcriptional changes with respect to the control condition and the WD and HT conditions (Figure 6B). The upregulation of *OsHSP70CP1* (0.9-fold WD + HT; 0.6-fold HT; 0.4-fold WD) and *OsHSP17.9A* (2.1-fold WD + HT; 0.6-fold HT; 0.4-fold WD) in primed plants under *a*[CO_2_] shows the effectiveness of WD at the vegetative stage, causing a priming effect in plants experiencing HT at the reproductive stage. The positive effect of *e*[CO_2_] promoting priming in rice spikelets is shown by the upregulation of *OsHSP16*.9*A* (6.7-fold WD + HT; 3.7-fold HT; 0.0-fold WD), *OsHSP70.1* (1.0-fold WD + HT; −0.8-fold HT; 2.4-fold WD), and *OsHSP70.6* (2.0-fold WD + HT; −1.2-fold HT; 1.4-fold WD) in primed spikelets with respect to the HT and WD conditions.

## 3. Discussion

### 3.1. Rice Yield Is Positively Affected by e[CO_2_]

We identified that increased CO_2_ concentration (*e*[CO_2_]) positively affected rice yield by increasing number of panicles per plant, thousand-grain weight, and yield, while decreasing panicle sterility (Figure 1). Our initial hypothesis that the increase in [CO_2_] could affect the development of cross-tolerance in rice plants was rejected, since in our experimental conditions of priming and [CO_2_], analyzing variables and sampling time, we do not detect interaction of *e*[CO_2_] and priming in rice yield. 

Panicle production is one of the components of rice yield that shows the greatest increase under *e*[CO_2_] [18], corroborating with previous reports [19], which showed a significant increase in the number of panicles in plants of three rice genotypes (Akita 63, Koshihikari, and Takanari) grown in *e*[CO_2_]. Increased rice yield due to *e*[CO_2_] was also reported as result of increased tillering [20]. We previously observed that BRS Querencia, the cultivar analyzed in this study, showed increased tillering at *e*[CO_2_] (unpublished data), which corroborates with the increased yield at *e*[CO_2_].

The physiological effects of *e*[CO_2_] in improving crop yield have been revealed. The increase in atmospheric [CO_2_] leads to a reduced wheat flower mortality, suggesting that flower viability is supported by the greater availability of photoassimilates translocated to floral organs [21]. Flower viability is highly associated with spikelet sterility, which was greatly lower in rice plants exposed to *e*[CO_2_] (Figure 1B). Therefore, the lower percentage of spikelet sterility in *e*[CO_2_] detected here may be an effect of the greater supply of photoassimilates to the reproductive organs during anthesis, since rice spikelet sterility may occur due to the low availability of photoassimilates during flowering, as reported in the literature [22]. *e*[CO_2_] helps to improve crop yield, as well as positively affecting TGW. A significant increase in TGW of2% was detected in rice grown in *e*[CO_2_] (Figure 1D). Our findings corroborate with previous studies that show that *e*[CO_2_] promotes an increase in grain weight [18,23]. The increase TGW of 1.3% was detected in *e*[CO_2_] was mainly attributed to the increase in the proportion of spikelets produced in the primary branches and in the source–sink ratio during panicle emission [24].

Interestingly, yield in indica rice cultivars is more positively affected by *e*[CO_2_] than japonica cultivars [18,25]. We previously observed that BRS Querencia, an indica Brazilian cultivar, has higher yield in *e*[CO_2_], mainly with respect to other indica Brazilian rice cultivars (unpublished data). Another important factor for increased rice yield is the proper nitrogen (N) supply, since it enhances *e*[CO_2_] effect [25]. Here, we performed fertilization at the important rice-developing stages—sowing, vegetative, and reproductive—which may have helped with the *e*[CO_2_] effect on rice yield. Proper N fertilization can also enhance photosynthetic pigment contents, leading to increased photosynthetic efficiency and in yield by the improved photochemical efficiency [26]. Here, a negative effect of *e*[CO_2_] was detected in chlorophyll *a* and *b* content with respect to *a*[CO_2_] (Figure 3D,E), even with the improvement in rice yield in *e*[CO_2_]. Our chlorophyll quantifications were performed in rice flag leaves after anthesis, which may show the positive effect of *e*[CO_2_] in increasing rice yield by translocating photoassimilates from flag leaves to grain filling.

Despite the increase in yield, rice plants grown in *e*[CO_2_] showed higher cellular damage revealed by the increased H_2_O_2_ content, even with the increase in the activity of the antioxidative enzymes CAT and SOD (Figure 3B,C). Increasing in SOD and CAT activity in response to *e*[CO_2_] were also observed in tomato (*Solanum lycopersicum*) and *Robinia pseudoacacia* [27,28], showing that it is a common response in plants facing *e*[CO_2_]. In addition, *e*[CO_2_] caused a decrease in chlorophyll content (Figure 3D–F), with it suggested that *e*[CO_2_] accelerates chlorophyll degradation in flag leaves, leading to senescence, indicating that after transition to the reproductive stage, there is a higher N translocation from flag leaves to the panicle to support grains development [29]. Furthermore, under *e*[CO_2_], there may be a limitation in N uptake caused by the decline in transpiration, and there may also occur an increase in the C:N ratio, causing a dilution of N concentration in the leaf, reducing the chlorophyll content [30,31]. It should be noted that in a previous study, it was found that the reduction in the chlorophyll content in rice plants does not seem to negatively affect grain productivity, and even favors an increase in the photosynthetic rate, resulting in greater photosynthetic efficiency of N use under conditions of high light intensity [32]. Overall, the dynamics in chlorophyll content as function of N or *e*[CO_2_] and its impact in rice yield is a complex issue that should be better understood. However, it is expected that the yield effect of *e*[CO_2_] offsets the negative effect in crop yields caused by the climate changes [18].

### 3.2. Priming with Water Deficit Increases Rice Yield

Agriculture is facing a challenge to produce resilient and climate-smart crops considering climate change and increased global food demand. Exploring plant stress memory has been shown to be a potential tool for crop improvement. Plant stress memory is associated with genetic and epigenetic regulation after the initial stress. Thus, pre-exposure to eliciting factors of the parents, grandparents, or plants at early developmental stages, a process called priming, can induce short-term or long-term stress memory that helps plants to tolerate additional stress in current or subsequent generations [33].

The priming effect has been reported as an improver of yield potential in many crops. Here, we identified that rice plants primed with water deficit at the vegetative stage were able to produce a greater number of grains per panicle and had increased yield (Figure 2A,C). Primed wheat plants with water deficit at the vegetative stage had lower losses of grain productivity after exposure to water deficit at the post-anthesis stage [34]. The physiology behind these profiles in primed wheat plants is associated with a better water status, caused by the increase in osmoprotector (proline and sucrose) content, and greater activity of the enzymatic antioxidant system (SOD, CAT, and APX) that acts in the reduction of oxidative damage on cell membranes through ROS scavenging [35]. Also, this priming mechanism enables greater efficiency of the electron transport chain of photosynthesis, as well as greater carboxylation activity of the RUBISCO enzyme. Using the same model, priming wheat plants with water deficit at the vegetative stage followed by water deficit at the post-anthesis stage reported an increased potential in grain development, which was associated with a high concentration of cytokinins, auxins, and gibberellins, as well as a low ABA content during the initial development of the grains [36]. This results in larger initial grain size and final grain filling, compared to plants that were not subjected to priming. 

The mechanisms activated by priming with water deficit are also important for the activation of thermotolerance in plants subjected to heat stress. In rice plants, for example, the development of thermotolerance involves the action of antioxidant enzymes (SOD, CAT, APX, and POD) due to the increased ROS accumulation and MDA content, together with the activity of heat shock proteins (HSP70) [2,37,38,39,40]. In our analysis, we observed changes in transcriptional regulation of *OsHSPs* (Figure 6B) and increased MDA content (Figure 4) related to the priming, but we did not detect the effect of water deficit, heat, or priming in the antioxidant enzymes SOD, CAT, and APX (Table 1). We standardized the heat stress to occur during rice anthesis, and our sampling for antioxidant enzymatic activity was performed right after the end of three days (72 h) under stress. We speculate that our sampling to analyze antioxidant enzymes activity was too early (72 h after heat stress) to be detected, since previous reports show increased total antioxidant capacity in experiments, with more the time of exposure of rice plants to heat from the heading stage until the end of the anthesis stage [41].

Here, we report that primed rice plants with water deficit conditions at the vegetative stage and subsequently submitted to heat stress at anthesis showed increased yield compared to water-deficit-stressed and heat-stressed plants (Figure 2C). Yield in primed plants had a 7% increase with respect to plants grown in control conditions, a 9% increase in yield with respect to heat-stressed plants, and an 11% increase in yield with respect to water deficit-stressed plants. Our results clearly show the positive effect of priming with water deficit at vegetative stage to improve yield in heat-stressed rice plants at reproductive stage. In addition, our priming strategy was able to lead rice plants to cross-tolerance. That effect was observed in wheat plants primed with water deficit during the stem elongation stage. Primed wheat plants were able to trigger cross-tolerance to heat stress due to heat at the grain filling stage, since an increase in carboxylation rate and a reduction in grain yield loss were detected compared to plants subjected to heat stress without priming [42].

### 3.3. Tolerance to Heat Stress Induced by Priming with Water Deficit

Cross-tolerance between stresses is a phenomenon that involves the synergistic co-activation of specific responsive pathways that allow for an increase in stress tolerance [43]. The expression profile of *OsHSPs* identified in rice spikelets shows the effect transcriptional memory by induction. *OsHSP16.9A*, *OsHSP70.1*, and *OsHSP70.6* were upregulated in primed plants that received *e*[CO_2_], showing transcriptional memory by induction (Figure 6B). HSPs are a highly conserved protein family and are divided into high-molecular-mass proteins (HSP100, HSP90, HSP70, and HSP60), small-molecular-mass proteins (HSP20), and small heat shock proteins (sHsps) [44]. Interestingly, *OsHSP70.1* and *OsHSP70.6* encode proteins from the same HSP70 family. The expression of *HSP70* gene family is highly variable when induced by heat stress since the promoters are structurally diverse with respect to their cis-acting elements’ constitution [45]. The expression of the *HSP70* genes can be induced by various abiotic stresses such as heat, cold, salinity, and water deficit, as well as by biotic stresses caused by fungi, bacteria, and viruses [46]. Also, HSP70s are also involved in tolerance to water deficit. Wheat tolerance to water deficit when primed with water deficit was shown to be regulated by HSP70s in a crosstalk regulation involving ABA and jasmonic acid [47]. Overall, here, we highlighted the importance of the activation of the *OsHSP70* gene family regulating cross-tolerance at the molecular level.

*OsHSP16.9A* encodes sHsps, proteins with molecular mass of 15 to 42 kDa. sHsps, including *OsHSP16.9A*, are associated with the ability of rice to avoid heat stress, since they are involved in a protein complex after being induced by heat stress functioning as molecular chaperones, and their expression levels are considered as markers of rice tolerance to heat stress [48]. Altogether, it corroborates with the priming effect observed in rice spikelets revealed by the highly detected upregulation of *OsHSP16.9A* in 50% in primed plants with respect to heat-stressed plants (Figure 6B). Acting as a marker of heat stress tolerance, the *OsHSP16.9A* expression profile detected in primed plants highlights the efficiency of the priming strategy used here and the cross-tolerance induced by water deficit.

Our findings associating HSP expression and cross-tolerance are supported by studies in other plant species. In *A. thaliana*, increased expression of the *AtHSP18.2* and *AtHSP21* activates a signal transduction network to regulate and induce heat stress memory [49]. Specifically, the expression of HSP70 was observed as a fundamental component for the development of cross-tolerance between arsenic and heat stress. In this case, the pre-exposure of rice seedlings to arsenic improved the plant’s ability to express *HSP70* efficiently during subsequent stress, indicating common aspects in the signal transduction cascade between different stresses [50]. Furthermore, it seems that post-transcriptional memory prolongation of some HSPs requires interaction with other cellular components. In *A. thaliana*, for example, the interaction between HSP101 and HSA32 (heat-stress-associated 32-kD) delays the degradation of HSP101, so this mechanism makes it possible to extend the memory of heat stress [51].

In fact, the molecular signaling involved in cross-tolerance has been revealed as a very complex and fine-tuned integrative process. In *Festuca arundinacea* plants, cross-tolerance between water deficit and heat stress is related to the phosphorylation of numerous proteins that have important metabolic functions, including proteins involved in pre-mRNA processing (RNA splicing), in transcription control, in addition to proteins of protection and signaling to stresses, such as HSPs [15]. The morphophysiological changes associated with cross-tolerance have been also understood. Wheat (*Triticum aestivum*) plants previously exposed to water deficit develop cross-tolerance to the second stress caused by heat, when they can present greater leaf water potential and chlorophyll content, in addition to maintaining a high photosynthetic rate during stress [16]. Priming induced by water deficit at the initial vegetative stage is also able to trigger the mechanism of cross-tolerance to cold stress in wheat stem elongation, mainly because the priming effect causes an increase in the activity of the enzymatic antioxidant system, protecting the photosynthetic apparatus against possible oxidative damage when facing the second stress [52]. Pre-exposure of *Poa pratensis* plants to water deficit also induces the ability to tolerate subsequent heat stress. In this case of cross-tolerance, an improvement in the performance of antioxidant enzymes (SOD, CAT, and APX) was observed in plants subjected to the first stress. Furthermore, the authors emphasize the important involvement of the HSP90 protein in protecting cellular components against damage caused by heat stress [17]. We did not observe a significant effect of priming, water deficit, or heat stress in the analyzed physiological parameters. We hypothesize that our sampling may have been collected too early to detect changes in chlorophyll content and antioxidant enzyme activity.

## 4. Materials and Methods

### 4.1. General Experimental Conditions

Rice plants were grown during their entire lifecycle in two different CO_2_ concentration regimes: current ambient CO_2_ concentration (*a*[CO_2_]) at 400 ± 40 μmol mol^−1^ and elevated CO_2_ concentration at 700 ± 40 μmol mol^−1^ *(e*[CO_2_]). The experiments were conducted in plant growth chambers (CCP 8410—Instalafrio^®^), with an internal area of 8.2 m^2^, equipped with LED light (containing red, blue, and green light), and with sensors that regulate temperature (Novus^®^ N1200; 5 to 50 °C +/− 1 °C), air humidity (Novus(R) RHT P10; 0–95% +/− 5%) and atmospheric CO_2_ control (Vaisala^®^ CO_2_ Probe GMP252; 0–3000 µmol mol^−1^ +/− 40 µmol mol^−1^). The CO_2_ sensor identifies changes in CO_2_ (Vaisala^®^ CO_2_ Probe GMP252), allowing the difference in CO_2_ concentration between chambers to be controlled with precision, 400 ± 40 μmol mol^−1^ (*a*[CO_2_]) or 700 ± 40 μmol mol^−1^ (*e*[CO_2_]).

The experiments were performed in 8 L plastic pots, filled with 7 kg of dry soil classified as planossolo háplico [53], from a rice field experimental area. According to physical–chemical analysis, the soil presented the following parameters: pH 5.3; calcium (2.4 cmolc dm^−3^); magnesium (1.4 cmolc dm^−3^); phosphorus (7.9 mg dm^−3^); potassium (24 mg dm^−3^); 0.83% organic matter, 18% clay; CTCpH 7 (7.1) and 57% base saturation. Eight rice seeds were sown per pot, and after seedling emergence, they were thinned to adjust the final population to five plants per pot.

Rice seeds of the cultivar BRS Querência were used. BRS Querência was significantly responsible for *e*[CO_2_] in our previous experiments (unpublished data). Plants were grown at a temperature of 28/25 °C (day/night) throughout the cycle, except during the period of heat stress for the group of plants subjected to heat (35/28 °C day/night, Table 2 and Figure 7). The relative humidity of the air in the chambers was set at 75% and the photoperiod was set to 14 h (14/1 0 h—day/night). During the experiments, the LED boards were kept at approximately 70 cm above the plant canopy, providing approximately 800 µmol m^−2^ s^−1^ of flux density of photosynthetically active photons. The fertilization was carried out in three stages, the first at the time of sowing, in which base fertilization was carried out with the application of 0.06 g of N, 0.26 g of P_2_O_5_, and 0.26 g of K_2_O per pot in the form of formulated fertilizer (5-20-20). At the (V4) stage, we applied 0.28 g of N per pot in the form of urea (45% N), and at stage (R1), we applied of 0.28 g of N per pot.

The experiments were carried out in a completely randomized experimental design, within a factorial arrangement, consisting of eight replications, each pot being considered as one repetition. Factor A consisted of the treatments of water deficit at the four-leaf stage (no-stress and drought stress), heat at anthesis (normal temperature, high temperature), and priming with water deficit at the four-leaf stage and heat stress at anthesis; and factor B was two treatments: [CO_2_] (*a*[CO_2_] = 400 ± 40 μmol mol^−1^ and *e*[CO_2_] = 700 ± 40 μmol mol^−1^). The nomenclature of each treatment combination is demonstrated in Table 2. At the anthesis, 10 panicles and the respective flag leaves were marked for sampling and evaluation.

### 4.2. Procedures for Establishing Water and Temperatures Stresses

The water conditions and temperatures imposed on the treatments are shown in Figure 7. The control water condition was characterized by the maintenance of soil moisture at pot capacity from sowing to the V5–6 stage. After this period, flood irrigation was imposed, with continuous maintenance of a 3 cm water slide until the culture matured. The water deficit was imposed by restricting irrigation from the V4 to the V5 stage. The water deficit stress condition was defined as the moment when the stressed plants presented a reduction of approximately 70% of the stomatal conductance compared to the irrigated plants (Figure 8A). At the end of the water deficit, plants were re-irrigated by flooding, as described above for the control treatment. Stomatal conductance was also monitored 24 and 48 h after rehydration to assess plant recovery. The monitoring of stomatal conductance was carried out using a leaf porometer (SC-1 Leaf Porometer—Decagon Device, Inc., Pullman, WA, USA). Readings were taken on the abaxial face of the median portion of the leaf blade of the last completely expanded leaf, and in each period, two plants per repetition were evaluated.

Regarding temperature conditions, for plants that did not undergo heat stress (CNT and WD), the optimal condition was characterized as 28/25 °C (day/night). Plants submitted to heat stress (HT and WD + HT) were grown under optimal temperature until the R4 stage. Upon reaching anthesis, these plants were transferred to a growth chamber with a temperature set at 35/28 °C (day/night), and the temperature transition was carried out through a temperature curve (Figure 8B) and maintained for 72 h. After this period, four replicates were again placed under optimal temperature conditions until the end of the cycle, and four replicates were used for sampling.

### 4.3. Evaluations of the Priming and CO_2_ Effects

Three days after anthesis (end of heat stress), plant material from four replications of each treatment was collected for biochemical analysis (flag leaves) and gene expression (flag leaves and spikelets). Samples were quickly placed in a thermal box containing liquid nitrogen and then stored in an ultrafreezer at −80 °C.

#### 4.3.1. Yield Components and Biomass Accumulation

At the rice maturation stage, panicles were collected, and the number of panicles per pot and grain yield per pot were determined using four replicates. From the 10 panicles marked when they reached anthesis, the number of full grains per panicle, the percentage of spikelet sterility, and the weight of one thousand grains were determined. Plant material from the shoot (leaves and stems) and from the root system were collected and dried in an oven with forced air circulation at a temperature set at 65 °C for 72 h. After drying, samples were weighed on an analytical balance to determine the shoot (SDW) and root (RDW) dry weight, with the results expressed in g per pot.

#### 4.3.2. Biochemical Analysis

##### Quantification of Hydrogen Peroxide Content and Lipid Peroxidation

The material for quantification of hydrogen peroxide and lipid peroxidation, using four replicates collected three days after anthesis (end of heat stress), was obtained from 250 mg of fresh plant material (flag leaves) that were macerated and homogenized in 2 mL of trichloroacetic acid (0.1% TCA) and centrifuged at 12,000× *g* for 20 min, at a temperature of 4 °C. The quantification of hydrogen peroxide (H_2_O_2_) was performed as described by Loreto and Velikova (2001), for which a reaction medium containing 10 mM potassium phosphate buffer (pH 7.0), 1 M potassium iodide, and sample extract was prepared; the reaction was kept at 30 °C for 10 min. The H_2_O_2_ was determined by reading the absorbance at 390 nm and the results expressed in nmol H_2_O_2_ g^−1^ of fresh mass.

Lipid peroxidation was determined by means of the quantification of thiobarbituric acid reactive species (TBARS) in four replicates collected three days after anthesis (end of heat stress), by the accumulation of malondialdehyde (MDA), as described by Heath and Packer (1968). TBARS were quantified by adding 300 μL of sample extract to the reaction medium containing 1.7 mL of thiobarbituric acid (TBA) (0.5% *m*/*v*) prepared in TCA (10% *m*/*v*). Afterwards, the reaction was kept in a water bath at 90 °C for 20 min, then the reaction was stopped by rapid cooling on ice for 10 min, and the absorbance was determined at 535 nm and 600 nm. The MDA content was calculated using the following equation: [MDA] = (A535–A600—Ablank)/(ξ.b), ξ meansthe extinction coefficient = 155 mM^−1^ cm^−1^ and b means the optical length = 1. Finally, lipid peroxidation was expressed in nmol of MDA g^−1^ of fresh mass.

##### Antioxidant Enzymes Activity

For the extraction of antioxidant enzymes, 250 mg of fresh plant material (flag leaves) were used in four replicates collected three days after anthesis (end of heat stress), which were macerated with polyvinylpolypyrrolidone (PVPP) (10% *w*/*w*) and homogenized in 2 mL of extraction buffer, composed of 100 mM potassium (pH 7.8), 0.1 mM EDTA (ethylenediaminetetraacetic acid), 10 mM ascorbic acid, and distilled water. Afterwards, the material was centrifuged at 12,000× *g* for 20 min at 4 °C. The supernatant was collected and used to determine the enzymatic activity of superoxide dismutase (SOD) (EC 1.15.1.1), catalase (CAT) (EC 1.11.1.6), and ascorbate peroxidase (APX) (EC 1.11.1.11) and for quantification of total proteins. The quantification of total proteins was performed by the method described by Bradford (1976), the absorbance was adjusted to 595 nm, and bovine serum albumin was used as standard.

SOD activity was evaluated by the enzyme’s capacity to inhibit the photoreduction of nitrotetrazolium blue (NBT) [54] in an incubation medium composed of 50 mM potassium phosphate buffer (pH 7.8), 14 mM methionine, EDTA 0.1 µM, NBT 75 µM, riboflavin 2 µM, distilled water, and enzymatic extract of the samples. In 96-well microplates, samples containing reaction medium and controls containing only reaction medium were illuminated for 10 min and placed in an adapted thermal box containing a 20 W fluorescent lamp, while the reference standard (white) was kept in the dark. Readings were taken at 560 nm. One SOD unit was defined as the amount of enzyme capable of inhibiting the photoreduction of NBT by 50% under the tested conditions, and the result was expressed in units min^−1^ mg of protein^−1^.

CAT activity was evaluated based on H_2_O_2_ consumption [55]. The reaction medium consisted of 100 mM potassium phosphate incubation buffer (pH 7.0), H_2_O_2_ (12.5 mM), distilled water, and enzymatic extract. CAT activity was quantified by the decrease in absorbance at 240 nm during a period of 3 min, using the H_2_O_2_ molar extinction coefficient (ε = 36 µM^−1^ cm^−1^), with the activity result expressed in μmol H_2_O_2_ min^−1^ mg protein^−1^.

The evaluation of the APX activity was carried out in relation to the ascorbate oxidation rate [56]. The reaction medium consisted of 37.5 mM potassium phosphate buffer (pH 7.0), ascorbic acid (0.25 mM), H_2_O_2_ (5 mM), distilled water, and the enzymatic extract. Ascorbate oxidation was monitored by the decrease in absorbance at 290 nm over 3 min, and the APX activity was calculated using molar extinction coefficient ε= 2.8 mM^−1^ cm^−1^. Results were expressed in μmol ASA min^−1^ mg protein^−1^.

##### Quantification of Chlorophyll *a*, Chlorophyll *b*, and Carotenoids

The pigments were quantified by the dimethylsulfoxide (DMSO) method [57,58]. For quantification, 20 mg of fresh mass was used in four replicates collected three days after anthesis (end of heat stress) and obtained through sections of leaf tissue approximately 1.5 cm long. To extract the pigments, 7 mL of DMSO was added to test tubes containing the leaf tissue. The tubes were kept in the dark at 65 °C for 120 min. Readings were performed at 480, 649, and 665 nm, and the quantifications of each pigment were calculated using the equations described by Wellburn [58]. The readings of all reactions and quantifications mentioned above were performed in a spectrophotometer with a microplate reader (Molecular Devices SpectraMax^®^).

#### 4.3.3. Gene Expression Analysis

##### RNA Extraction and cDNA Synthesis

Total RNA was extracted from 100 mg of plant material (flag leaves and spikelets) using PureLink Plant RNA (Invitrogen^®^) according to the manufacturer’s recommendations in three replicates collected three days after anthesis (end of heat stress). RNA concentration and quality were evaluated by spectrophotometry using NanoVue^TM^ (GE Healthcare). Only samples with a 260/280 ratio (indication of protein contamination) and 260/230 ratio (indication of reagent contamination) between 1.8 and 2.2 were used. The integrity of the RNA samples was checked using agarose gel electrophoresis. To remove possible contamination with genomic DNA, 1 µg of RNA was treated with DNase I (Invitrogen^®^) before reverse transcription. Each sample was converted to cDNA using oligo(dT) and SuperScript™ III First-Strand Synthesis System kit (Invitrogen), according to the manufacturer’s recommendations.

##### Reverse Transcription–Quantitative Polymerase Chain Reaction (RT-qPCR)

Quantification of gene expression in RT-qPCR was performed according to the MIQE Guidelines [59], using oligonucleotides for target genes and reference genes (Table 3). To determine the amplification efficiency and specificity of each oligonucleotide, validation experiments were performed for each oligonucleotide using four cDNA dilutions. Oligonucleotides that showed efficiency between 90 and 110% and a single peak in the dissociation curve were used for quantification analyzes of gene expression. Gene expression experiments were performed in a LightCycler^®^ 480 Instrument II thermal cycler (Roche) with three biological replicates and three technical replicates. Reactions were performed containing 1 µL of cDNA at a 1:25 dilution (determined in the validation experiment), 11.0 µL of UltraPure™ DNase/RNase-Free Distilled Water (Invitrogen^®^), 0.25 µL of ROX Reference Dye (Invitrogen^®^), 2.0 µL 10× PCR Buffer (Invitrogen^®^), 1.5 µL 50 mM Magnesium Chloride (Invitrogen^®^), 0.05 µL Platinum™ Taq DNA Polymerase, 0.2 µL dNTPs, 3.0 µL SYBR Green I (Invitrogen^®^), and 1.5 µL of each oligonucleotide (forward and reverse) in a reaction of 20 µL final volume. Reactions without cDNA were also performed for each pair of oligonucleotides and used as a negative control. PCR reactions were performed under the following conditions: initial denaturation at 95 °C for 5 min; 45 cycles of 95 °C for 20 s, 60 °C for 15 s, and 72 °C for 20 s in LightCycler^®^ 480 Multiwell Plates 96 (Roche). The melting curve was performed under the following conditions: 95 °C for 5 s, 70 °C for 1 min, and a temperature ramp up to 95 °C, increasing by 0.5 °C for 5 s.

The quantification of the gene expression of genes coding for *OsHSPs* was calculated using the ΔΔCT method [60]. To verify the effect of the [CO_2_], each treatment control condition, HT, WD, and WD + HT, in the *e*[CO_2_] condition, was normalized in relation to its respective treatment (HT, WD, or WD + HT) in the *a*[CO_2_] condition. To verify the effect of priming, each stress condition, HT, WD, and WD + HT, at each *a*[CO_2_] or *e*[CO_2_], was normalized in relation to control condition at *a*[CO_2_] concentration. Normalizations were performed for each tissue (flag leaf and spikelet) and in relation to the reference genes *Os18S*, *OsEF1α*, and *OsUBQ5* [61].

**Table 3 plants-12-03133-t003:** Rice (*Oryza sativa* L.) oligonucleotides used for analysis in RT-qPCR reactions of *OsHSP* gene expression.

Gene	ID	Oligonucleotide—(5′-3′)		References
*SMALL HEAT-SHOCK PROTEIN 16.9A*	*OsHSP16.9A*	Forward	CATTCTCCCTCGACCTCTGG	[48]
		Reverse	GGTGCCACTTGTCGTTCTTGT	
*SMALL HEAT-SHOCK PROTEIN 17.9A*	*OsHSP17.9A*	Forward	GGCAGCATCTTCCCGTCCTT	[48]
		Reverse	CGTCGTCCACCTCCACCTTG	
*SMALL HEAT-SHOCK PROTEIN 20*	*OsHSP20*	Forward	AAGTTCCTCCGCAGGTTCC	[62]
		Reverse	GAGCACGCCGTTCTCCAT	
*SMALL HEAT-SHOCK PROTEIN 24.15*	*OsHSP24.15*	Forward	GATCAAGGCGGAGATGAAGAAC	[63]
		Reverse	ACTCGACGTTGACCTGGAAGA	
*HEAT-SHOCK PROTEIN 70CP1*	*OsHSP70CP1*	Forward	CCAGGCGATGTGAAGGGTAA	[64]
		Reverse	CAGCATCGATGACATCACCT	
*HEAT-SHOCK PROTEIN 70.1*	*OsHSP70.1*	Forward	TCGAGCTCTCCGGCATCCCT	[65]
		Reverse	TCACGTAGGACCACCACCAGCT	
*HEAT-SHOCK PROTEIN 70.6*	*OsHSP70.6*	Forward	GATCAACGTCTGCTTCGACA	[65]
		Reverse	AGTCGACCTCCTCGATCTTG	
*18S ribosomal RNA*	*Os18S*	Forward	CTACGTCCCTGCCCTTTGTACA	[61]
		Reverse	ACACTTCACCGGACCATTCAA	
*Elongation Factor 1-α*	*OsEF1α*	Forward	TTTCACTCTTGGTGTGAAGCAGAT	[61]
		Reverse	GACTTCCTTCACGATTTCATCGTAA	
*Ubiquitin 5*	*OsUBQ5*	Forward	ACCACTTCGACCGCCACTACT	[61]
		Reverse	ACGCCTAAGCCTGCTGGTT	

##### Statistical Analysis

Data for spikelet sterility (%) were transformed into $xT=\sqrt{x+1}$. All the data were submitted to analysis of variance by F test with 5% significance level. Analyses were performed using the statistical software R (www.r-project.org/, accessed on 30 April 2021). Data from treatments at each CO_2_ concentration were analyzed by *t*-test (*p* ≤ 0.05). To verify the effect of cross-tolerance, data from the treatments CNT (control condition), WD (water deficit), HT (heat), and WD + HT (priming) were submitted to means comparison analysis through Tukey’s honestly significant difference test.

## 5. Conclusions

Water deficit at the vegetative stage (V4) is an environmental trigger that can be used to increase rice yield and increase the number of grains per panicle. The increase in CO_2_ concentration in 700 ppm from seed germination to anthesis does not affect the priming with water deficit at the vegetative stage. Our findings are limited to a controlled environment, and further analysis must be performed to bring this to the field.

## Figures and Tables

**Figure 1 plants-12-03133-f001:**
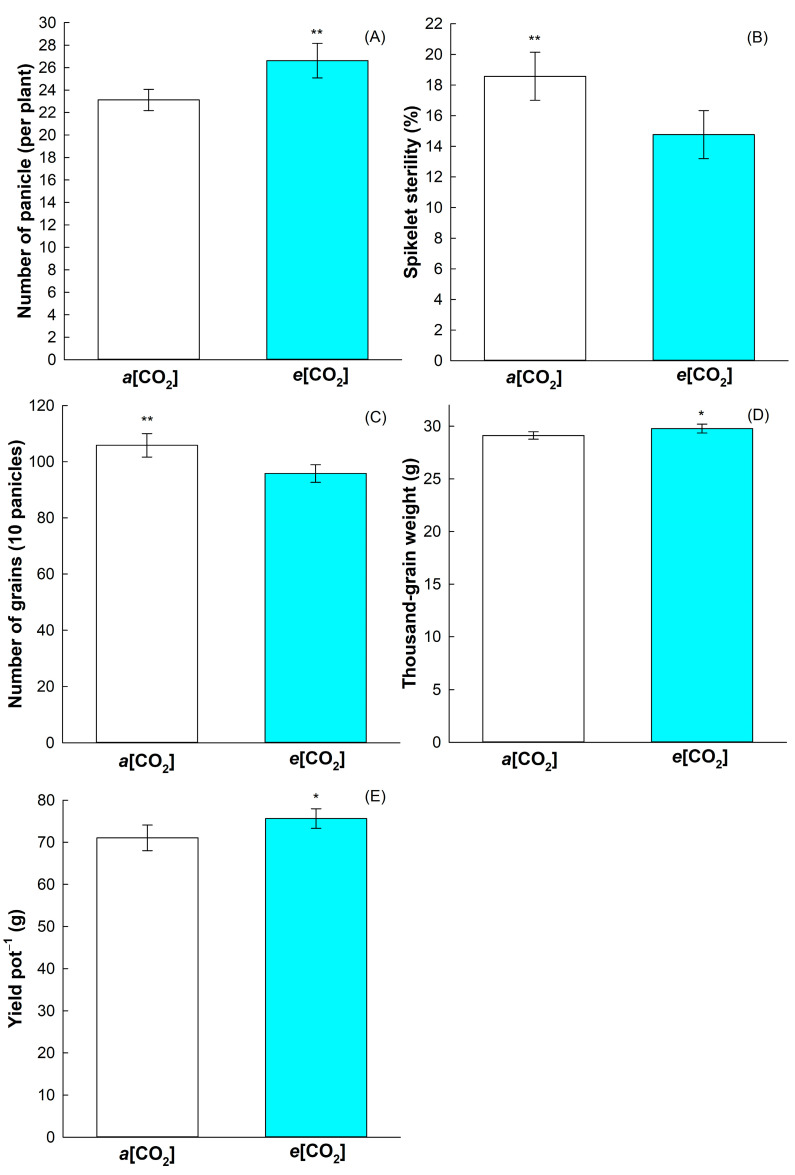
Effect of CO_2_ concentrations (*a*[CO_2_] = 400 ± 40 μmol mol^−1^ and *e*[CO_2_] = 700 ± 40 μmol mol^−1^) in the number of panicles (**A**), spikelet sterility % (**B**), number of grains (**C**), weight of thousand grain (**D**), and yield (**E**) of BRS Querência rice cultivar. Error bars correspond to confidence interval, *n* = 4. The two-tailed Student’s *t*-test was used for statistical analyses (** indicates *p*-value < 0.01 and * indicates *p*-value < 0.05).

**Figure 2 plants-12-03133-f002:**
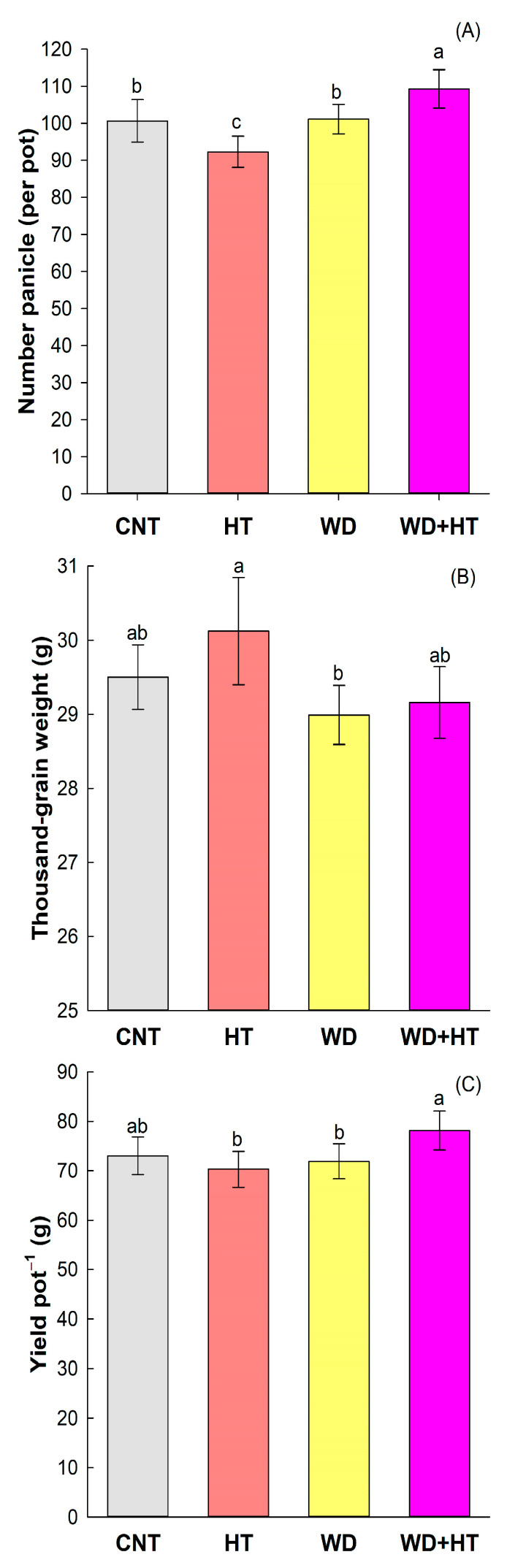
Effect of the priming and stress conditions in the number of grains per panicles (**A**), thousand-grain weight (**B**), and yield (**C**) of BRS Querência rice cultivar. CNT—control condition, HT—heat stress at anthesis (35/28 °C day/night at R4 stage), WD—water deficit at V4–V5 stage, and WD + HT—primed plants with water deficit at V4–V5 stage and heat stress at anthesis (35/28 °C day/night at R4 stage). Error bars correspond to confidence interval, *n* = 4. Lowercase letters indicate the results of Tukey’s test comparing the growing conditions CNT, HT, WD, and WD + HT (*p* ≤ 0.05).

**Figure 3 plants-12-03133-f003:**
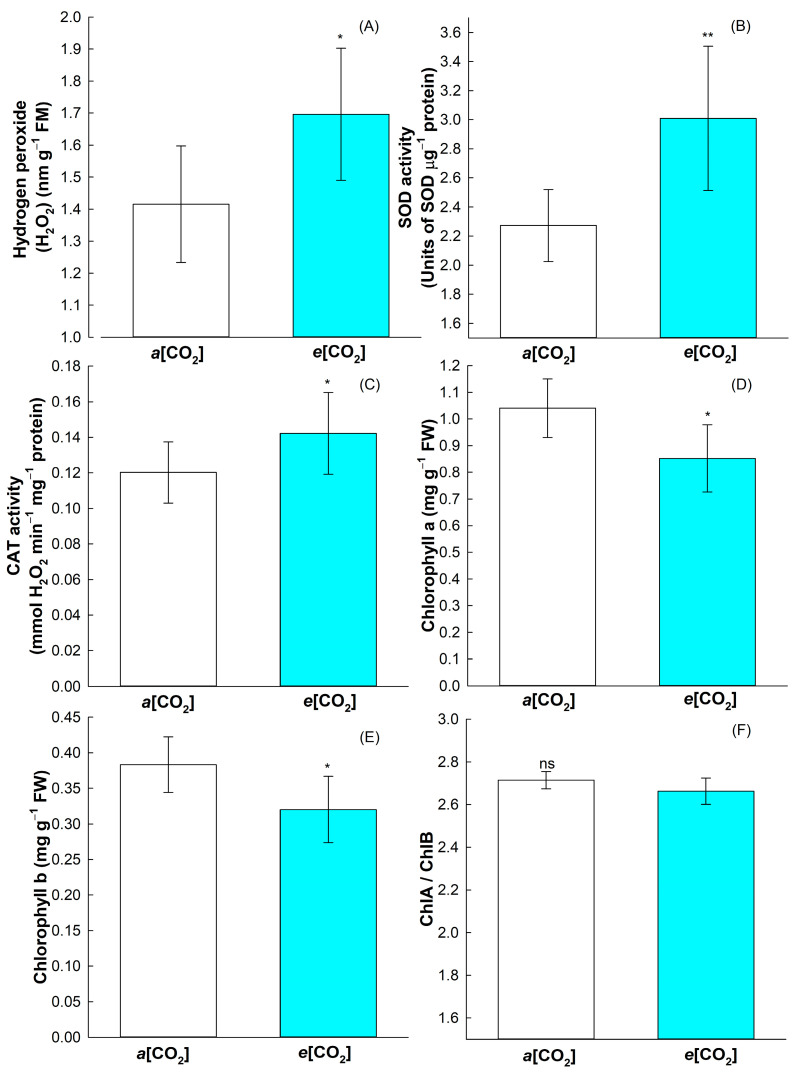
Effect of CO_2_ concentrations (*a*[CO_2_] = 400 ± 40 μmol mol^−1^ and *e*[CO_2_] = 700 ± 40 μmol mol^−1^) in the hydrogen peroxide content (**A**), superoxide dismutase (SOD) activity (**B**), catalase (CAT) activity (**C**), chlorophyll *a* content (**D**), chlorophyll *b* content (**E**) and chlorophyll *a* and chlorophyll *b* ratio (**F**) of BRS Querência rice cultivar. Error bars correspond to confidence interval, *n* = 4. The two-tailed Student’s *t*-test was used for statistical analyses (** indicates *p*-value < 0.01 and * indicates *p*-value < 0.05). ns: non-significant.

**Figure 4 plants-12-03133-f004:**
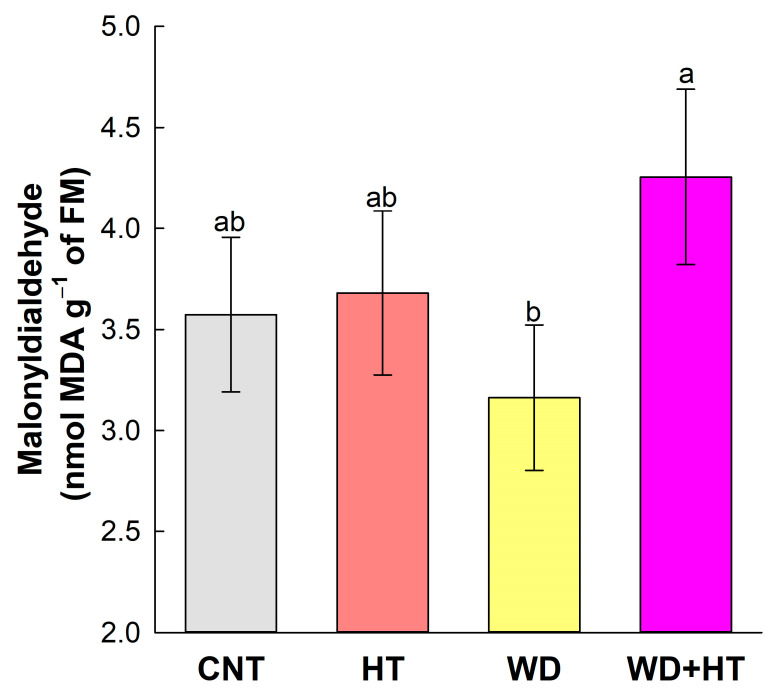
Effect of the priming and stress conditions in Malonyldialdehyde (MDA) content of BRS Querência rice cultivar. CNT—control condition, HT—heat stress at anthesis (35/28 °C day/night at R4 stage), WD—water deficit at V4–V5 stage and WD + HT—primed plants with water deficit at V4-V5 stage and heat stress at anthesis (35/28 °C day/night at R4 stage). Error bars correspond to confidence interval, *n* = 4. Lowercase letters indicate the results of Tukey’s test comparing treatments (*p* ≤ 0.05).

**Figure 5 plants-12-03133-f005:**
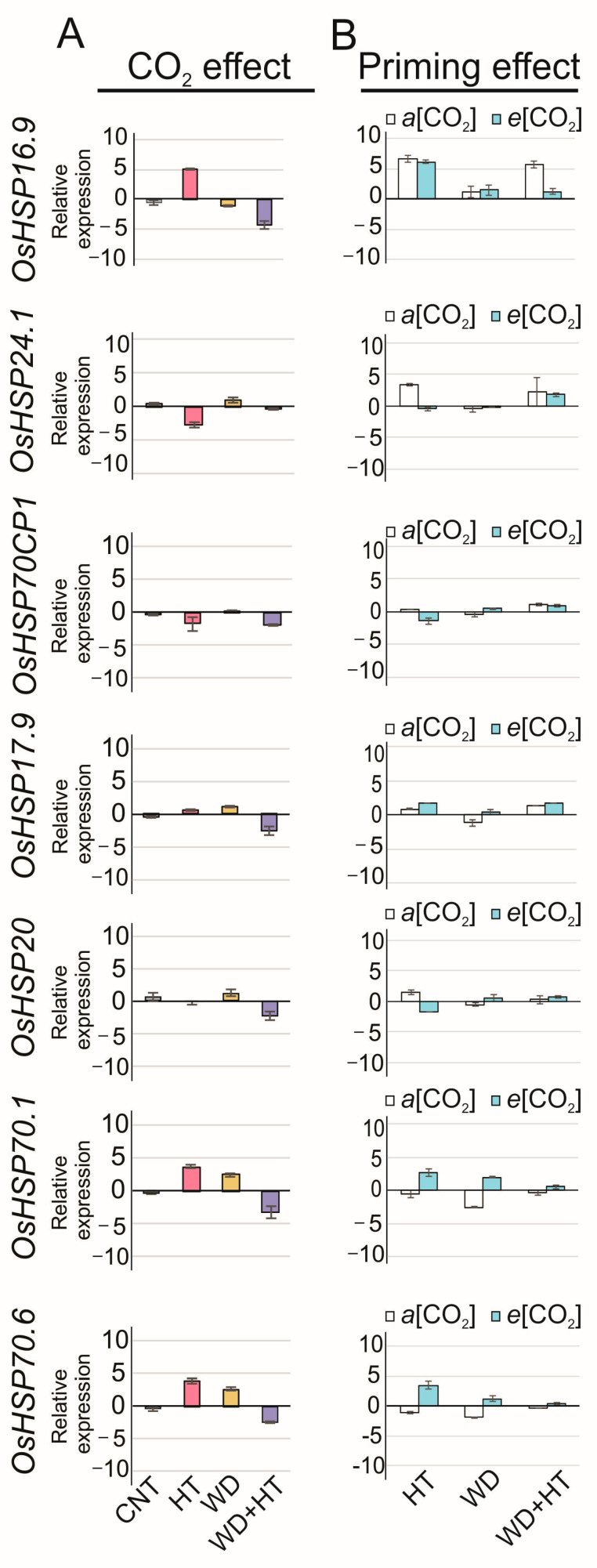
Effect of CO_2_ concentration ([CO_2_]) and priming on the relative expression of genes encoding for heat shock proteins (HSPs) in rice flag leaves. The rice plants were cultivated in two [CO_2_] regimes {400 ± 40 ppm of CO_2_ (*a*[CO_2_]) and 700 ± 40 ppm of CO_2_ (*e*[CO_2_])}, subjected to control condition (CNT), heat stress at anthesis (HT), water deficit at the V4–V5 stage (WD), and plants that receive priming with water deficit at the V4–V5 stage and heat stress at anthesis (WD + HT). For the effect of CO_2_ concentration, each condition, CNT, HT, WD, and WD + HT, at *e*[CO_2_], was normalized relative to its respective condition at *a*[CO_2_]. For the effect of priming, each stress condition, HT, WD, and WD + HT, at each CO_2_ concentration, *a*[CO_2_] or *e*[CO_2_], was normalized with respect to CNT at *a*[CO_2_]. Error bars correspond to standard error, *n* = 3.

**Figure 6 plants-12-03133-f006:**
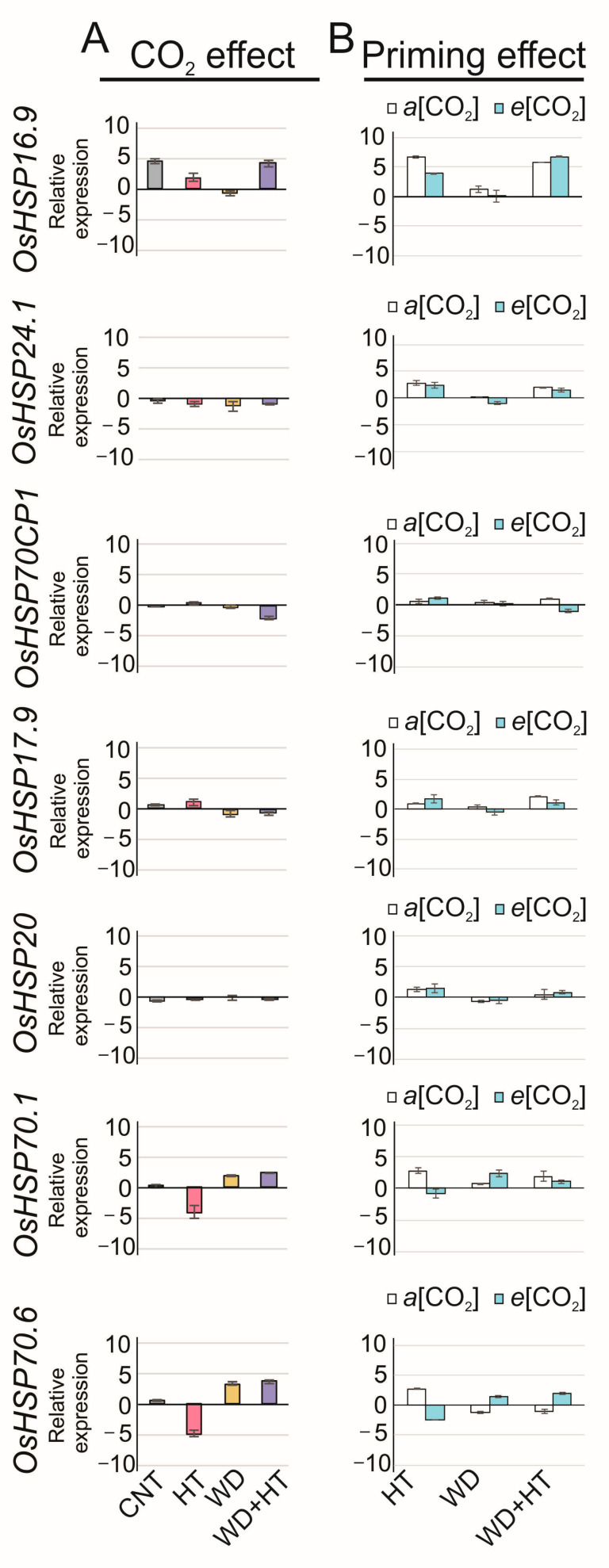
Effect of CO_2_ concentration ([CO_2_]) and priming on the relative expression of genes coding for heat shock proteins (HSPs) in rice spikelets. The rice plants were cultivated in two [CO_2_] regimes, 400 ppm of CO_2_ (*a*[CO_2_]) and 700 ppm of CO_2_ (*e*[CO_2_]), subjected to control condition (CNT), heat stress at anthesis (HT), water deficit at the V4–V5 stage (WD), and plants that receive priming with water deficit at the V4–V5 stage and heat stress at anthesis (WD + HT). A: For the effect of CO_2_ concentration, each condition, CNT, HT, WD, and WD + HT, at *e*[CO_2_], was normalized in relation to its respective condition at *a*[CO_2_]. B: For the effect of priming, each stress condition, HT, WD, and WD + HT at each CO_2_ concentration, *a*[CO_2_] or *e*[CO_2_], was normalized with respect to CNT at *a*[CO_2_]. Error bars correspond to standard error, *n* = 3.

**Figure 7 plants-12-03133-f007:**
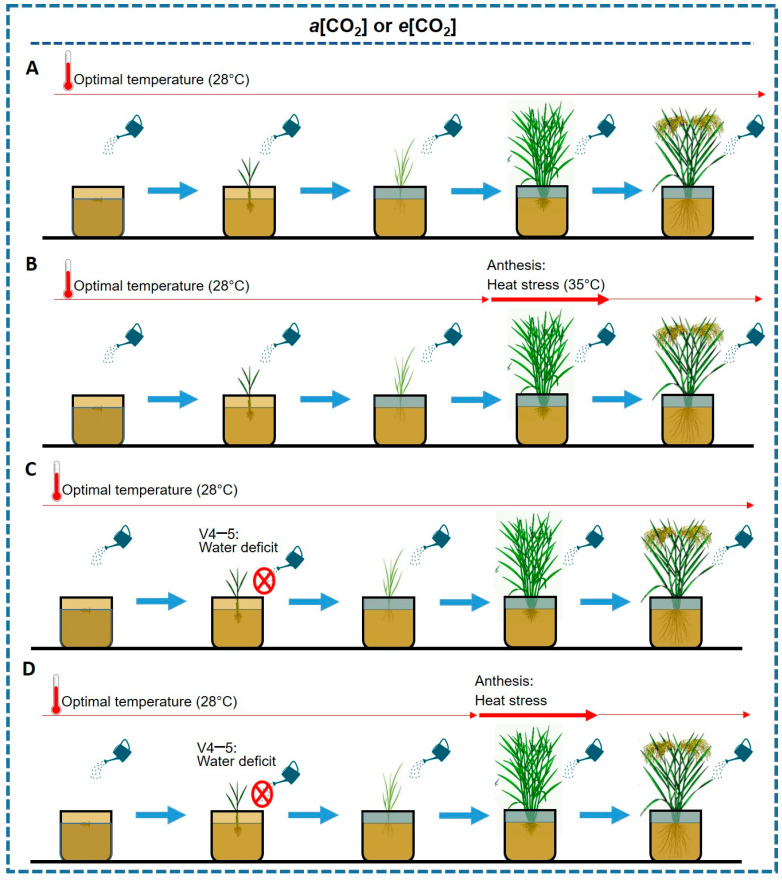
Graphic representation of water regimes and temperatures of the treatments: Control condition (**A**), heat (**B**), water deficit—WD (**C**), and heat plus water deficit—heat + WD (**D**).

**Figure 8 plants-12-03133-f008:**
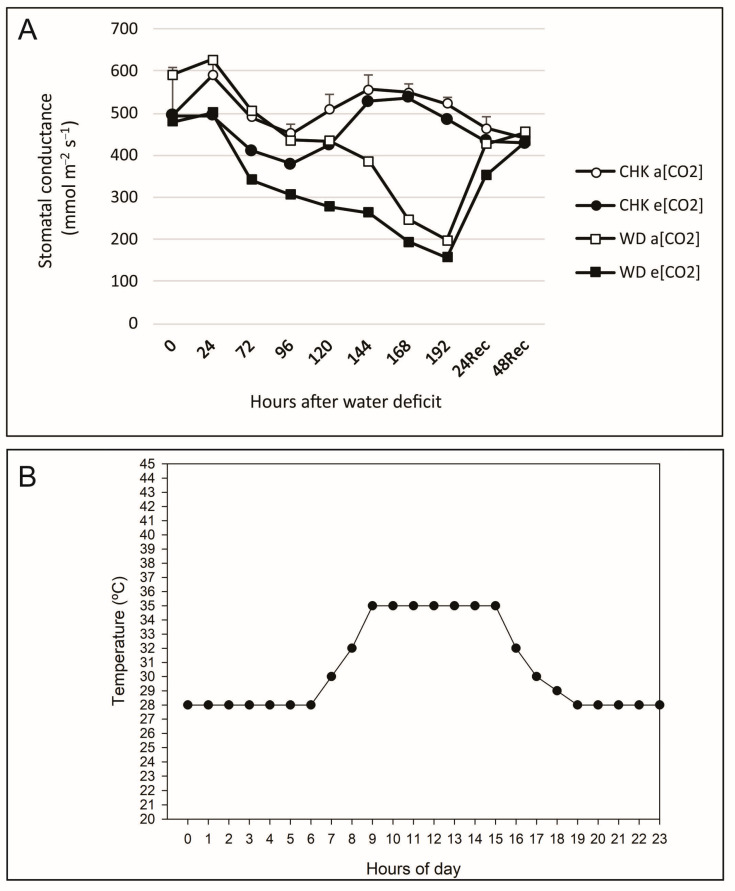
Experimental conditions. (**A**) Stomatal conductance (gs) of rice plants at the V4–5 stage grown in irrigated water conditions at 400 and 700 ppm of atmospheric CO_2_ (control condition), and plants submitted to water deficit (WD) at 400 and 700 ppm of CO_2_. Vertical lines at each point indicate confidence intervals (95%); and (**B**) daily temperature ramp configuration for heat stress treatment.

**Table 1 plants-12-03133-t001:** *p*-value of the simple effects and the interaction between the factors CO_2_ concentration ([CO_2_]) and stress caused by water deficit and heat stress and priming.

Variable	[CO_2_]	Stresses	[CO_2_] x Treatments
SDW	0.058	0.930	0.065
RDW	0.141	0.05	0.684
Panicles vase^−1^	0.001 **	0.596	0.682
Grains panicle^−1^	0.000 **	0.000 **	0.467
Sterility (%)	0.004 **	0.821	0.642
TGW	0.017 *	0.022*	0.937
Yield pot^−1^	0.016 *	0.27*	0.669
H_2_O_2_	0.010 **	0.451	0.491
MDA	0.468	0.019	0.675
SOD	0.001 **	0.192	0.976
CAT	0.026 *	0.064	0.164
APX	0.083	0.588	0.780
Chlo a	0.004 **	0.562	0.221
Chlo b	0.007 **	0.518	0.212
Chlo T	0.004 **	0.551	0.219
Carot	0.102	0.363	0.077

SDW: shoot dry weight; RDW: root dry weight; TGW: thousand-grain weight; H_2_O_2_: hydrogen peroxide; MDA: malonyldialdehyde; SOD: superoxide dismutase; CAT: catalase; APX: ascorbate peroxidase; Chlo *a*: chlorophyll *a*; Chlo *b*: chlorophyll *b*; Chlo T: total chlorophyll; Carot: carotenoids. * *p* ≤ 0.05; ** *p* ≤ 0.01.

**Table 2 plants-12-03133-t002:** Description of treatments applied in cross-stress experiments.

Treatments	Description
CNT *a*[CO_2_]	Control condition—400 ppm of CO_2_; irrigated and 28/25 °C day/night (optimal conditions)
HT *a*[CO_2_]	Heat—400 ppm of CO_2_; irrigated and with heat stress at R4 stage (35/28 °C day/night)
WD *a*[CO_2_]	Water deficit—400 ppm of CO_2_; water deficit at V4–5 stage and optimal temperature
WD + HT *a*[CO_2_]	Priming—400 ppm of CO_2_; water deficit at V4–5 stage and heat stress at R4 stage (35/28 °C day/night)
CNT *e*[CO_2_]	Control condition—700 ppm of CO_2_; irrigated and optimal temperature
HT *e*[CO_2_]	Heat—700 ppm of CO_2_; irrigated and with heat stress at R4 stage (35/28 °C day/night)
WD *e*[CO_2_]	Water deficit—700 ppm of CO_2_; water deficit at V4–5 stage and optimal temperature
WD + HT *e*[CO_2_]	Priming—700 ppm of CO_2_; water deficit at V4–5 stage and heat stress at R4 stage (35/28 °C day/night)

## Data Availability

The datasets used and/or analyzed during the current study are available from the corresponding author on reasonable request.

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
