# Peer review of "Water Deficit at Vegetative Stage Induces Tolerance to High Temperature during Anthesis in Rice"

_plants, 2023, doi:10.3390/plants12173133_

Round 1
Reviewer 1 Report
This study is interesting and novel, however, there are several issues that should be addressed before it can be considered for publication. The study is related to priming using different stress factors, but it is not always clear if CO2 treatments were done on primed and unprimed plants. It was written that author aim "to verify the possible development of cross-tolerance between stresses", however, as far as I understand, Fig. 1 and 3 have only CO2 levels as treatments with no priming. In the legend of each figure I found only referral to "BRS Querência rice cultivar" with no mentioning of priming treatments. Moreover, it seems like some of the results related to CO2 experiments are not part of the general goal of this study and are not emphasized in the title of the manuscript. Thus, the connection between CO2 to priming should be clarified.
For gene expression, it seems that the effect of eCO2 concentration for each condition was normalized relatively to the respective condition at aCO2. One might wonder, why the authors choose to do so, presenting all results in comparison to the house keeping genes for each CO2 level may have been better to present these results. Generally, the figure for gene expression are visually small and a scale of 5 units is very narrow to show the actual differences. Also, scale should be adjusted to show the differences equally among all data units (-10-10).
For all experiments, the number of replicates is not always clear, I suggest to clarify it through the text in the Materials and Methods section for each experiment. This is also true for the time point were samples for each analysis were collected (gene expression, enzyme activity, etc.), please specify. This is a critical issue since comparison to other studies related to these reactions may be impaired if they have been taken in very different time points. Authors should refer to this fact in the discussion section.
English should be improved, maybe a native English speaker can have a look and make some editing.
Specific comments
Presenting bars may impair the ability of the reader to understand the actual differences among treatments, thus, box-plot are better in this case.
For each figure please insert in the legend the number of replicates, n=?.
Figure 2: yield per plot is in g and not Kg? The visual of the figure itself look a bit stretched.
83-84: since the material and methods section appears bellow, acronyms should be given in full at first mentioning according to their order in the text.
108: here and in other places across the manuscript, control plants and not check.
136: authors should refer to the lack of statistical differences, "however, the were not statistically significant then…"
246-248: this is very speculative; it should be phrased more carefully since authors have not tested it directly.
262, 275: which and not with.
232-288: as I mentioned in general comments section, experiments related to CO2 alone seem separate from the main goal of the study. Authors should better clarify the connection between the CO2 experiments to the rest of the study, especially in the discussion.
314: grain and not drain.
316-318: how this fact is connected to the current study?
326-327: can you cite other studies that have analyzed antioxidant activity at different time point? Why did select the specific time point of 72h after treatment?
372-381: it is not clear to me if this can be called memory, since the observed reaction is after exposure to the stress at early growth stage, these cases may be of different nature.
403: "may have been…"
405: redundant due.
418: delete the word previously.
435: what do you mean by "topdressing"?
618-626: the whole section seems very vague and should be rewritten to refer to the actual results and their conclusions. Also, here is the first mentioning of weed control in the entire manuscript, if it is important it should be discussed previously and also mentioned in the introduction section.
Should be improved.
Author Response
Reviewer 1 Comments
Reviewer comment: This study is interesting and novel, however, there are several issues that should be addressed before it can be considered for publication. The study is related to priming using different stress factors, but it is not always clear if CO2 treatments were done on primed and unprimed plants.
Authors response: Dear reviewer thank you for your comment. We improved the material and methods and the experiment scheme to clarify that plants from all treatments, water deficit, heat and priming were grown in both CO2 regimes, current ambient CO2 concentration and elevated CO2 concentration.
Reviewer comment: It was written that author aim "to verify the possible development of cross-tolerance between stresses", however, as far as I understand, Fig. 1 and 3 have only CO2 levels as treatments with no priming.
Authors response: Dear reviewer thank you for your comment. We do not detect CO2 effect on priming, which was revealed by the ANOVA. Taking it into account, we showed separated priming and CO2 effects. In addition, we concluded that increasing in CO2 do not affect cross tolerance development in our conditions and that the priming strategy was efficient to develop cross-tolerance since we observed improvement in some yield components and in the HSPs expression.
Reviewer comment: In the legend of each figure I found only referral to "BRS Querência rice cultivar" with no mentioning of priming treatments.
Authors response: Dear reviewer thank you for your comment. As showed by the ANOVA (table 1), we do not detect effect of the CO2 concentration in cross-tolerance development (priming) and we showed its simple effects (CO2 or priming) separated from each other. Taking it into account, we described and refer to the priming condition just in the figures we show priming effect (Figure 2 and Figure 4).
Reviewer comment: Moreover, it seems like some of the results related to CO2 experiments are not part of the general goal of this study and are not emphasized in the title of the manuscript. Thus, the connection between CO2 to priming should be clarified.
Authors response: Dear reviewer thank you for your comment. We opt to do not emphasize the CO2 in the title since we do not find effect of the CO2 on priming. We improved the text to highlight that CO2 did not affect the priming effect in our experimental conditions.
Reviewer comment: For gene expression, it seems that the effect of eCO2 concentration for each condition was normalized relatively to the respective condition at aCO2. One might wonder, why the authors choose to do so, presenting all results in comparison to the house keeping genes for each CO2 level may have been better to present these results. Generally, the figure for gene expression are visually small and a scale of 5 units is very narrow to show the actual differences. Also, scale should be adjusted to show the differences equally among all data units (-10-10).
Authors response: Dear reviewer thank you for your comment. We used the delta delta Ct method to normalize all the gene expression assays. In this method, we always used a mean of three housekeeping genes, in our case we used 18S ribosomal RNA, Elongation Factor 1-α and Ubiquitin 5, and the target genes that in our case were gene encoding Heat Shock Proteins. Also, we need to use an experimental condition as base line and we choose a[CO2], so that each variation observed in e[CO2] is translated in changes with respect to a[CO2]. Is that possible to use just the delta Ct to show the effect of the target gene with respect only to the housekeeping genes. However, to standardize our method among all our gene expression assays we opted to show the delta delta Ct and also because using the a[CO2] as baseline shows the difference of e[CO2].
Reviewer comment: For all experiments, the number of replicates is not always clear, I suggest to clarify it through the text in the Materials and Methods section for each experiment. This is also true for the time point were samples for each analysis were collected (gene expression, enzyme activity, etc.), please specify. This is a critical issue since comparison to other studies related to these reactions may be impaired if they have been taken in very different time points. Authors should refer to this fact in the discussion section.
Authors response: Dear reviewer thank you for your comment. We improved figure legends, material and methods and included the replicate information in all of this section as suggested.
Reviewer comment: English should be improved, maybe a native English speaker can have a look and make some editing.
Authors response: Dear reviewer thank you for your comment. We improved English grammar.
Specific comments
Reviewer comment: Presenting bars may impair the ability of the reader to understand the actual differences among treatments, thus, box-plot are better in this case.
Authors response: Dear reviewer thank you for your comment. We opt to use presenting the data in bars since we show the mean and confidence interval that shows de differences among the treatments.
Reviewer comment: For each figure please insert in the legend the number of replicates, n=?.
Authors response: Dear reviewer thank you for your comment. We inserted the number of replicates in all figure legends.
Reviewer comment: Figure 2: yield per plot is in g and not Kg? The visual of the figure itself look a bit stretched.
Authors response: Dear reviewer thank you for your comment. Yes, we measured the grams of seeds per pot. We improved figure quality.
Reviewer comment: 83-84: since the material and methods section appears bellow, acronyms should be given in full at first mentioning according to their order in the text.
Authors response: Dear reviewer thank you for your comment. We inserted the acronyms should in full in the first time it appears in the text.
Reviewer comment: 108: here and in other places across the manuscript, control plants and not check.
Authors response: Dear reviewer thank you for your comment. We change from check to control along the text.
Reviewer comment: 136: authors should refer to the lack of statistical differences, "however, the were not statistically significant then…"
Authors response: Dear reviewer thank you for your comment. We inserted a statement related to the lack of differences.
Reviewer comment: 246-248: this is very speculative; it should be phrased more carefully since authors have not tested it directly.
Authors response: Dear reviewer thank you for your comment. We rephrased the sentence and stated that it was observed in other studies available in the literature.
Reviewer comment: 262, 275: which and not with.
Authors response: Dear reviewer thank you for your comment. We change from “with” to “which” in both, line 262 and line 275.
Reviewer comment: 232-288: as I mentioned in general comments section, experiments related to CO2 alone seem separate from the main goal of the study. Authors should better clarify the connection between the CO2 experiments to the rest of the study, especially in the discussion.
Authors response: Dear reviewer thank you for your comment. We improved the writing in the begging of the discussion section, stated that we do not detected interaction of the increased CO2 concentration and priming effects in our experimental conditions and highlighted that our hypothesis that increasing in CO2 concentration affect cross-tolerance was rejected.
Reviewer comment: 314: grain and not drain.
Authors response: Dear reviewer thank you for your comment. We corrected the word grain instead of drain.
Reviewer comment: 316-318: how this fact is connected to the current study?
Authors response: Dear reviewer thank you for your comment. We remove this sentence from the text.
Reviewer comment: 326-327: can you cite other studies that have analyzed antioxidant activity at different time point? Why did select the specific time point of 72h after treatment?
Authors response: Dear reviewer thank you for your comment. We analyzed antioxidant activity at 72 hs after the heat stress because it was the right time of the end of the heat stress that was applied only during rice anthesis. We included other report related the antioxidant enzymatic activity in other times of sampling.
Reviewer comment: 372-381: it is not clear to me if this can be called memory, since the observed reaction is after exposure to the stress at early growth stage, these cases may be of different nature.
Authors response: Dear reviewer thank you for your comment. We changed the term “memory” to cross-tolerance.
Reviewer comment: 403: "may have been…"
Authors response: Dear reviewer thank you for your comment. We change the sentence as suggested.
Reviewer comment: 405: redundant due.
Authors response: Dear reviewer thank you for your comment. We removed the redundant sentence.
Reviewer comment: 418: delete the word previously.
Authors response: Dear reviewer thank you for your comment. We change the sentence as suggested.
Reviewer comment: 435: what do you mean by "topdressing"?
Authors response: Dear reviewer thank you for your comment. We change the sentence to a most proper writing.
Reviewer comment: 618-626: the whole section seems very vague and should be rewritten to refer to the actual results and their conclusions. Also, here is the first mentioning of weed control in the entire manuscript, if it is important it should be discussed previously and also mentioned in the introduction section.
Authors response: Dear reviewer thank you for your comment. We improved the conclusion section to summarize our findings.

Reviewer 2 Report
Please find the attached file to get the minor comments about the paper.
I am curious to know what made you look at carbon dioxide concentration. Is there any practical implication or particular significance of knowing the performance of the rice plants related to carbon dioxide concentration? If yes, that point can be included in the paper.

Only minor changes are required.
Author Response
Reviewer 2 Comments
Reviewer comment - Line 81: Need to clarify which groups are being compared.
Authors response: Dear reviewer thank you for your comment. We included the information of the factors have been compared as suggested.
Reviewer comment – Line 83-84: Define WD and HT treatment. People will read "result" section before "method".
Authors response: Dear reviewer thank you for your comment. We inserted WD and HT definition.
Reviewer comment – Line 98: What is e[Co2], it should be clarified somewhere.
Authors response: Dear reviewer thank you for your comment. We included what e[Co2] and a[Co2] means at the first time it appears in the text and in material and methods section.
Reviewer comment – Line 104-105: Typo of high and higher.
Authors response: Dear reviewer thank you for your comment. We corrected the sentence keeping just the word “higher”.
Reviewer comment – Line 104-105: What is CHK?
Authors response: Dear reviewer thank you for your comment. CHK is the check condition, but is was changed to control condition as suggested by the reviewer 1.
Reviewer comment – Line 183: The picture is too small.
Authors response: Dear reviewer thank you for your comment. We reorganized the figure from a horizontal to a vertical arrangement.
Reviewer comment – Line 623: Check grammar.
Authors response: Dear reviewer thank you for your comment. We improved the sentence.
